# BDQ: Bidirectional Diagonal Quantization for LLMs

## Abstract

Post-training quantization has emerged as a widely adopted technique for compressing and accelerating the inference of Large Language Models (LLMs). The primary challenges in LLMs quantization stem from activation outliers, which significantly degrade model performance especially at lower bit precision. While recent approaches attempt to mitigate outliers through linear transformations across feature dimensions, our analysis reveals that the transformed weights and activations still exhibit persistent outlier patterns with concentrated magnitude distributions. In this paper, we first model the mathematical relationship between quantization error and outliers, and then introduce a new metric Flatness to quantify the distribution of outliers. Based on this, we derive the theoretical optimal solution with respect to Flatness. Building on these insights, we propose Bidirectional Diagonal Quantization (BDQ), a novel post-training quantization framework that effectively disperses outlier patterns through optimized matrix transformations. BDQ strategically distributes outlier magnitudes across matrix dimensions via learned diagonal operations. Extensive experiments demonstrate that BDQ establishes a new quantization benchmark. It achieves less than 1% accuracy drop in W4A4 quantization on the LLaMA-3-8B model. In the more challenging W2A4KV16 experiment, compared to state-of-the-art approaches, BDQ reduces the performance gap by 39.1% on the DeepSeek-R1-Distill-LLaMA-70B model.

## 1 Introduction

Recent Large Language Models (LLMs) have achieved superior performance in multiple natural language processing tasks as their parameters grow (Yang et al., 2024; Grattafiori et al., 2024). However, increasing the scale of the parameters leads to significant increases in computational and storage costs (Xiao et al., 2023). Therefore, the efficient deployment of low-cost LLMs has become an urgent research direction (Ashkboos et al., 2025). Previous research can be divided into architecture-changing and architecture-preserving techniques.

Architecture-changing methods such as distillation (Han et al., 2015; Chen et al., 2020) and pruning (Zhu et al., 2024) reduce the size of the model by transferring knowledge or removing unimportant parameters, but require significant data and computation, making them impractical for LLMs. In contrast, architecture-preserving methods such as quantization (Frantar et al., 2022) and low-rank decomposition (Yuan et al., 2023) keep the model structure; quantization lowers weight precision, while low-rank methods approximate weight matrices. Quantization is especially popular in LLM deployment due to its efficiency and strong performance.

Post-Training Quantization (PTQ) has become a widely adopted technique for compressing and accelerating LLMs. During quantization, as shown in Figure 1a, outliers in the original data present huge challenges because the limited quantization space cannot adequately express the original data space, with most data accumulating in a few regions. Recent research has adopted linear transformations to address these challenges. The rotation transformation (Ashkboos et al., 2025; Liu et al., 2024) alleviates this phenomenon in Figure 1b. However, due to the presence of outliers, most of the data still accumulates in the Blue region. Existing methods are heuristic and haven't established direct mathematical relationships between outliers and quantization errors, nor optimized the distribution of the entire quantization space.

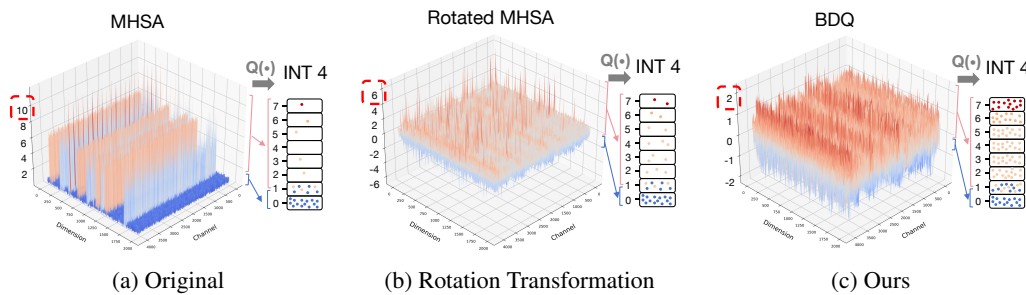

(a) Original      (b) Rotation Transformation      (c) Ours

Fig. 1: Activation distributions under different transformations for LLaMA3-8B. After quantization, values from various ranges are mapped to corresponding integer levels. The number of points within each box reflects the frequency of quantized values. A more uniform distribution of points indicates higher quantization quality. Blue dots represent values near zero, Orange dots indicate mid-range values, and Red dots correspond to large-magnitude values.

In this paper, we first establish the mathematical relationship between outliers and quantization errors, demonstrating that outliers influence quantization error at the quadratic level. Furthermore, we introduce the concept of **Flatness** as an effective indicator for quantifying the distribution of outliers. Inspired by Information-Entropy (Tsai et al., 2008), we define **Flatness** as evaluating each element's flatness in its row and column, extending to all elements in the matrix. Importantly, through mathematical derivation, we discovered the optimal solution for improving Flatness and demonstrated excellent advantages compared to previous methods, laying the foundation for developing more effective quantization methods.

Based on the above findings, we propose the **B**idirectional **D**iagonal **Q**uantization (**BDQ**) method. BDQ allocates two learnable diagonal transformation pairs for each fully connected layer in LLMs, applying simultaneous row-wise and column-wise scaling to redistribute outliers along both dimensions. We theoretically demonstrate that this formulation can achieve the optimal solution with respect to Flatness. In addition, a Hadamard orthogonal transformation is employed to further disperse outliers across the entire matrix. Meanwhile, it is widely known that only a small calibration set is utilized (e.g., 128 samples) during the quantization process, which can cause the model to overfit to a limited set of features, an effect shown experimentally to hinder outlier mitigation. To address this, BDQ introduces a Recursive Cross-Entropy loss that captures the state from previous iterations, thereby reducing overfitting and improving generalization. BDQ is a highly effective PTQ method for LLMs, consistently outperforming existing techniques across various models and benchmarks. In the W4A4KV4 setting, BDQ maintains over **99.1%** of full-precision accuracy. Furthermore, in the W2A4KV16 setting, BDQ reduces the performance gap of DeepSeek-R1-Distill-LLaMA-70B (Guo et al., 2025) by **39.1%** compared to the latest methods.

To our knowledge, we are the first to model the mathematical relationship between outliers and quantization errors, discovering that outliers are key factors affecting quantization accuracy. Meanwhile, we propose the **Flatness** metric reflecting the presence of outliers in the model and provide the optimal solution through mathematical derivation. The contributions of this work are summarized as follows:

- We first model the mathematical relationship between outliers and quantization errors, discovering that outliers are key factors that affect quantization accuracy.

- To quantify the outlier distribution, we propose the **Flatness** metric and provide the optimal solution through mathematical derivation. Compared to previous methods, this optimal solution demonstrates significant advantages.

- We propose a Bidirectional Diagonal Quantization (BDQ) that effectively reduces quantization errors. Extensive experiments show that BDQ significantly outperforms existing quantization methods.

## 2 RELATED WORK

### 2.1 ARCHITECTURE-CHANGING METHODS

Recent model compression research has focused on structural modifications to reduce complexity and size. Pruning techniques have progressed from early weight pruning (Han et al., 2015) to dynamic strategies that remove unimportant parameters during training (Chen et al., 2020), and to neural architecture search-based methods for optimal network structures (Zhang et al., 2021). Knowledge distillation has also advanced, from foundational teacher-student frameworks (Kim & Rush, 2016) to approaches combining self-supervised learning (Yang et al., 2022) and multi-modal distillation for preserving semantics (Zhao et al., 2024). However, these methods often entail high computational costs and slow processing, limiting their practical deployment.

### 2.2 ARCHITECTURE-PRESERVING METHODS

Post-training quantization (PTQ) is popular in LLMs for its efficiency, with methods mainly divided into weight-only and weight-activation quantization. FWSVD (Hsu et al.) and ASVD (Yuan et al., 2023) assess parameter or channel importance, while GPTQ (Frantar et al., 2022) and AWQ (Lin et al., 2024; Lee et al., 2023) reduce quantization error and address activation outliers. QuIP (Chee et al., 2023), QuIP# (Tseng et al., 2024), SmoothQuant (Xiao et al., 2023), and OmniQuant (Shao et al., 2023) further improve quantization with various techniques. I-LLM (Hu et al., 2024) supports integer-only inference, QuaRot (Ashkboos et al., 2025) uses random rotations, and SpinQuant learns rotations for 4-bit quantization (Liu et al., 2024). Quantization stands out over low-rank decomposition for its high accuracy and low cost.

## 3 MOTIVATION

In the model quantization process, let the weight or activation be $W \in \mathbb{R}^{m \times n}$, and assume the outlier value $|w_{\text{outlier}}| \gg \mathbb{E}[|W|]$, where $\mathbb{E}[|W|]$ represents the statistical expectation of the elements. The quantization process is determined by the scale $\triangle \in \mathbb{R}^+$ and the zero point $z \in \mathbb{Z}$, mapping floating-point values to the integer space as follows:

$$Q(w) = \text{round}\left(\frac{w}{\triangle}\right) + z, \triangle = \frac{\max(|w|)}{2^b - 1} \tag{1}$$

where $w$ is the original weight and $Q(w) - z \in \{0, 1, \ldots, 2^b - 1\}$ is the integer value after $b$-bit quantization. We set $x$ is the input of matrix, the quantization error is defined as:

$$\epsilon = wx - w'x \tag{2}$$

### 3.1 THE QUANTIZATION ERROR OF SINGLE OUTLIER

When the outlier value is included, let $\triangle$ be the selected scale factor and $b$-bit integer points. Assume the quantization range is set to $[-c, c]$:

$$\triangle = \frac{c}{2^b - 1} \tag{3}$$

If $|w_{\text{outlier}}|$ is large, then the adjustment leads to:

$$\triangle' = \frac{|w_{\text{outlier}}|}{2^b - 1} \tag{4}$$

Meanwhile, let $w_{\text{outlier}}$ represent the quantization bin, $\Delta = \frac{c}{2^b - 1}$ expands to $\frac{|w_{\text{outlier}}|}{2^b - 1}$. For any non outlier $w_i \in [-c, c]$, their upper limit of quantization error increases from $\frac{\Delta}{2}$ to $\frac{\Delta'}{2}$, that is:

$$|\epsilon_i| \leq \frac{\Delta x}{2} \xrightarrow{outlier} |\epsilon_i| \leq \frac{|w_{\text{outlier}}|x}{2^b - 1} \tag{5}$$

When $|w_{\text{outlier}}| \gg c$, the quantization error due to outliers can be significant. There is a proportional relationship between quantization error $\epsilon_i$ and outliers $w_{\text{outlier}}$.

## 3.2 THE QUANTIZATION ERROR OF ENTIRE MATRIX

The quantization error of the statistics and the characteristics of the weight can be assumed to follow a normal distribution $N(0, k^2\sigma^2)$ (where $k \gg 1$) (Ashkboos et al., 2025). The total quantization error can be expressed as:

$$E[\epsilon^2] = \frac{x}{mn} \sum_m^{j=1} \sum_n^{i=1} (w_{ij} - w'_{ij})^2 = (1-p)E[\epsilon^2_{\text{normal}}]x + pE[\epsilon^2_{\text{outlier}}]x \tag{6}$$

where $(1-p)E[\epsilon^2_{\text{normal}}]$ is Normal Contributions and $pE[\epsilon^2_{\text{outlier}}]$ is Outlier Contributions, $p$ is a coefficient related to the number of outliers. Due to the outlier value, as the scale factor $\triangle'$ increases, the variance of the normal term changes to:

$$E[\epsilon^2_{\text{normal}}] \approx \frac{\Delta'^2}{12} = \frac{k^2\sigma^2}{12(2^b - 1)^2} \tag{7}$$

And the mean error of the outlier itself, due to being truncated to the boundary of the quantization range, the error is:

$$E[\epsilon_{\text{outlier}}] = w_{\text{outlier}} - \text{sign}(w_{\text{outlier}}) \cdot (2^b - 1)\triangle' \tag{8}$$

when $|w_{\text{outlier}}| > (2^b - 1)\triangle'$, the $sign(\cdot)$ is a sign function. The average squared error is given by:

$$E[\epsilon^2_{\text{outlier}}] = (|w_{\text{outlier}} - (2^b - 1)\triangle'|)^2 \tag{9}$$

When the outlier value is significantly larger than the quantization range (i.e., $|w_{\text{outlier}}| \gg (2^b - 1)\triangle'$), outliers dominate the total quantification error (where $E[\epsilon^2_{\text{outlier}}] \gg E[\epsilon^2_{\text{normal}}]$), at this point:

$$E[\epsilon^2] \approx p \cdot w^2_{\text{outlier}}x \tag{10}$$

The total quantification error $E[\epsilon^2]$ and outliers $w_{\text{outlier}}$ exhibit a square relationship.

## 4 THE OPTIMAL SOLUTION FOR FLATNESS

In model quantization and compression, the original weight or activation matrix $W \in \mathbb{R}^{m \times n}$ often contains a few extremely large values that significantly exceed the magnitude of other elements. We refer to these values as outliers. The presence of outliers reduces the distinguishability of full-precision values within the limited quantization space, resulting in increased quantization error—one of the core challenges in quantization research. Existing studies primarily focus on mitigating outliers through scaling or linear transformations, and have achieved promising results. However, there remains a lack of a unified metric to evaluate the flatness of a matrix, making it difficult to assess or determine an optimal transformation strategy.

### 4.1 FLATNESS OF MATRIX

In information theory, entropy quantifies the uncertainty or randomness associated with a random variable or probability distribution. Higher entropy indicates greater uncertainty, lower information content, and a flatter probability distribution $P(x_i)$. The information entropy is defined as follows:

$$H(X) = -\sum_{i=1}^{z} P(x_i) \log P(x_i) \tag{11}$$

Inspired by Information-Entropy (Tsai et al., 2008), we propose an evaluation metric called **Flatness**, which quantifies the uniformity of the data distribution across the entire matrix. In this context, the elements of the matrix $W$ are treated as a part of probability values similar to $P(x_i)$ in the information entropy formulation. Importantly, the outliers in $W$ are distributed across different rows and columns of the model, so the flatness metric needs to ensure that the distributions of the rows and columns containing outliers are properly evaluated, the expression $\frac{W_{ij}^2}{\alpha_i \beta_j}$ can be similar to $P(x_i)$ in Eq. 11. Specifically, Flatness is formalized as:

$$F = \sum_{i=1}^{m} \sum_{j=1}^{n} \left( \frac{W_{ij}^2}{\alpha_i \beta_j} \ln \frac{W_{ij}^2}{\alpha_i \beta_j} \right) \tag{12}$$

where $\alpha_i > 0$ is the energy weight factor for the $i$-th row, $\beta_j > 0$ is the energy weight factor for the $j$-th column. The objective is to minimize the combined dispersion $F$, subject to the energy constraint:

$$\min_{\alpha_i, \beta_j} F \quad \text{s.t.} \quad \sum_{i,j} p_{ij} = \sum_{i,j} \frac{W_{ij}^2}{\alpha_i \beta_j} = 1 \tag{13}$$

Additional energy constraints (avoiding trivial solutions):

$$\sum_{i,j} \alpha_i W_{ij}^2 \beta_j = C, (C > 0) \tag{14}$$

We consider $\frac{W_{ij}^2}{\alpha_i \beta_j}$ as a probability distribution from two perspectives. Non-negativity and normalization: $W_{ij}^2 \geq 0, \alpha_i > 0, \beta_j > 0$, thus $p_{ij} = \frac{W_{ij}^2}{\alpha_i \beta_j} \geq 0$. The constraint $\sum_{i,j} \frac{W_{ij}^2}{\alpha_i \beta_j} = 1$ ensures that $\sum_{i,j} p_{ij} = 1$. This condition defines the distribution of probabilities. Information entropy: The information $H(p) = -\sum_{i,j} p_{ij} \ln p_{ij}$ measures the uncertainty of the distribution. As $p_{ij}$ increases, $H(p)$ becomes larger. In this problem, we hope to maximize $H(p)$ (i.e., maximize entropy), thereby making $p_{ij}$ approach the distribution of probabilities. The quality of this is the maximum entropy of the distribution, which is achieved by optimizing $p_{ij}$ as much as possible. The formula is $F = \sum_{i,j} \frac{W_{ij}^2}{\alpha_i \beta_j} \ln \left( \frac{W_{ij}^2}{\alpha_i \beta_j} \right) = \sum_{i,j} p_{ij} \ln p_{ij}$.

Additionally, we consider the constraints from two perspectives. Summary of Requirements: The information required is to ensure that the probability distribution $p_{ij}$ satisfies $\sum p_{ij} = 1$. If we hope to set $p_{ij} = \frac{W_{ij}^2}{\alpha_i \beta_j}$ as the probability distribution, then it must satisfy the condition that the total sums to 1. $\sum_{i,j} \frac{W_{ij}^2}{\alpha_i \beta_j} = 1$. Directly ensuring the summary requirement, guarantees $\sum p_{ij} = 1$ satisfies the summary condition. Physical Meaning: This constraint ensures that the total energy corresponding to the variable $W$ is $\sum W_{ij}^2$, while the roles of parameters $\alpha_i$ and $\beta_j$ are to redistribute the energy, making the distribution more uniform. The additional energy constraint $\sum \alpha_i W_{ij}^2 \beta_j = C$ is utilized to control the degree of bias in the release of factors, avoiding $\alpha_i, \beta_j \to 0$ or $\infty$ in the solution.

## 4.2 Finding the Optimal Solution

Introducing the Lagrange multiplier $\lambda$, the Lagrangian is constructed as:

$$\mathcal{L} = \sum_{i,j} \left( \frac{W_{ij}^2}{\alpha_i \beta_j} \ln \frac{W_{ij}^2}{\alpha_i \beta_j} \right) + \lambda_1 \left( 1 - \sum_{i,j} \frac{W_{ij}^2}{\alpha_i \beta_j} \right) + \lambda_2 \left( C - \sum_{i,j} \alpha_i W_{ij}^2 \beta_j \right). \tag{15}$$

We conducted derivation of optimality conditions. Taking partial derivatives with respect to $\alpha_k$ and $\beta_l$, and setting them to zero:

With respect to $\alpha_k$:

$$\frac{\partial \mathcal{L}}{\partial \alpha_k} = -\sum_j \frac{W_{kj}^2}{\alpha_k^2 \beta_j} \left( \ln \frac{W_{kj}^2}{\alpha_k \beta_j} + 1 + \lambda_1 \right) - \lambda_2 \sum_j W_{kj}^2 \beta_j = 0. \tag{16}$$

Reorganizing:

$$\sum_j \frac{W_{kj}^2}{\alpha_k^2 \beta_j} \left( \ln \frac{W_{kj}^2}{\alpha_k \beta_j} + 1 + \lambda_1 \right) = -\lambda_2 \sum_j W_{kj}^2 \beta_j. \tag{17}$$

Similarly, with respect to $\beta_l$:

$$\sum_i \frac{W_{il}^2}{\alpha_i \beta_l^2} \left( \ln \frac{W_{il}^2}{\alpha_i \beta_l} + 1 + \lambda_1 \right) = -\lambda_2 \sum_i \alpha_i W_{il}^2. \tag{18}$$

### 4.3 STRUCTURAL ANALYSIS OF THE SOLUTION

Utilizing the above formulas, we have clarified Row Independence and Column Independence. Row Independence refers to each $\alpha_k$ in the optimization process is determined only by the data in row $k$. Column Independence refers to each $\beta_l$ in the optimization process is determined only by the data in column $l$. This implies, $\alpha_i$ is a function of the data in row $i$, independent of other rows. $\beta_j$ is a function of the data in column $j$, independent of other columns.

Therefore, the optimal solution must be that $\alpha_i$ is given by a function of the data in row $i$, and $\beta_j$ is given by a function of the data in column $j$. By defining diagonal matrices $d_1 = \mathrm{diag}(\sqrt{\alpha_i})$ and $d_2 = \mathrm{diag}(\sqrt{\beta_j})$, we obtain: $V = d_1 W d_2$ as the unique optimal form. Notably, $V$ not only represents the theoretical optimal solution with respect to Flatness but also, according to Eq. 10, is the optimal form for reducing quantization error.

## 5 METHOD

Based on the theoretical solution $V$ obtained in Section 4, we propose Bidirectional Diagonal Quantization (BDQ) along with Recursive Cross-Entropy loss, which together theoretically yield the optimal Flatness of the matrix.

### 5.1 BIDIRECTIONAL DIAGONAL QUANTIZATION

We propose Bidirectional Diagonal Quantization (BDQ), a novel framework designed to mitigate the impact of outliers and enhance quantization performance. The key idea behind BDQ is to distribute the burden of outlier elimination across the entire matrix, as detailed in Section 3.

As illustrated in Figure 2, BDQ applies multiple transformation pairs both within and across LLM blocks globally. Specifically, based on the transformer architecture, each block learns four equivalent transformation pairs, with each pair consisting of two learnable diagonal matrices and one learnable rotation matrix. These transformations collaboratively reshape the distribution of weights and activations, making them more amenable to quantization. BDQ preserves equivalent transformations at the global network level. Therefore, when quantization is not applied, the network's output remains identical to that of the original model. More details are provided in the Appendix C.

We define the equivalent transformation pair as $E$, where $E$ consists of two diagonal matrices $< \Lambda_1, \Lambda_2 >$ and a rotation matrix $R$. Therefore, the forward inference process $y = xW$ is reformulated:

$$y = Q(\Lambda_1 x \Lambda_2 R) \cdot Q(R^T \Lambda_2^{-1} W \Lambda_1^{-1}) \tag{19}$$

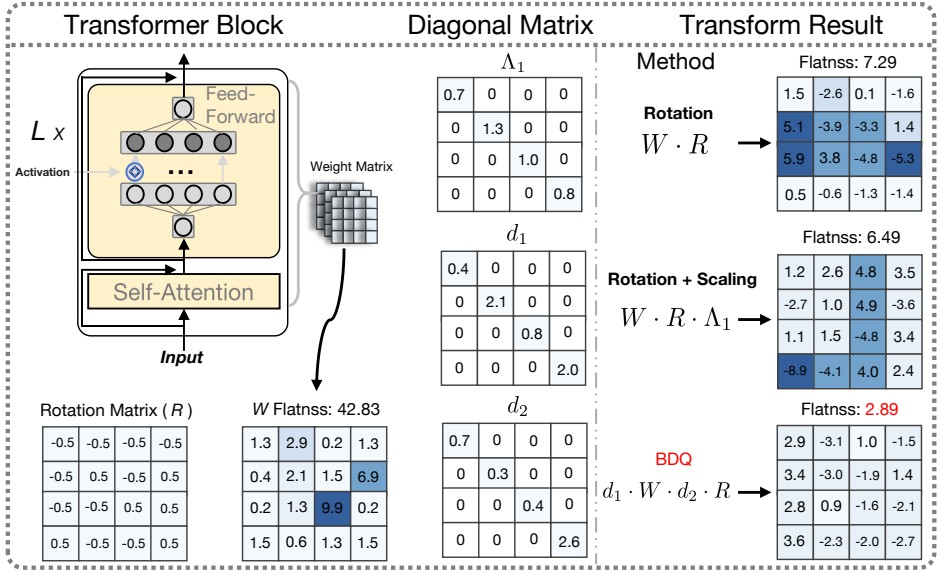

Fig. 2: The transformation results of different methods. The Rotation Matrix is a learnable random Hadamard matrix. The diagonal matrix is obtained by optimizing and converging utilizing deep neural networks.

| Alignment Dataset | Step | PPL (↓) | Arc-Easy (↑) | Arc-Challenge (↑) | Hellaswag (↑) | Flatnss (↓) |
|---|---|---|---|---|---|---|
| **Wikitext2** | 100 | 5.67 | **76.37** | **48.35** | **77.83** | **173.27** |
| | 200 | 5.24 | 76.26 | 48.30 | 77.76 | 184.37 |
| | 300 | 4.97 | 76.20 | 48.19 | 77.68 | 207.59 |
| | 400 | **4.80** | 76.07 | 48.07 | 77.65 | 232.89 |
| **C4** | 100 | 7.46 | 76.27 | 48.27 | **77.80** | 189.66 |
| | 200 | 6.83 | **76.32** | **48.36** | 77.78 | **180.53** |
| | 300 | 6.75 | 76.24 | 48.17 | 77.68 | 189.37 |
| | 400 | **6.61** | 76.15 | 48.09 | 77.59 | 203.99 |

Table 1: Experimental results of overfitting phenomenon on LLaMA3-8B .

where $Q(\cdot)$ represents the quantization function and $W$ refers to weight or activation. $\Lambda$ is a diagonal matrix, and the inverse of the diagonal elements of $\Lambda$ is obtained as $\Lambda^{-1}$. The rotation matrix $R$ is composed of a Hadamard matrix and an additional orthogonal matrix. Appendix A provides a detailed theoretical comparison with previous rotation-based methods, and the results demonstrate that our method has significant advantages in complexity and outlier elimination.

The optimization objective for the entire network can be formalized as follows:

$$\arg \min_{\Lambda_i, R_i} \mathcal{L}(\hat{y}, y; \Lambda_i, R_i, \theta) \tag{20}$$

The $\mathcal{L}(\hat{y}, y)$ represents the loss between the quantized network output $\hat{y}$ and the full-precision network output $y$. The $\theta$ denotes the parameters of the frozen network.

## 5.2 RECURSIVE CROSS-ENTROPY LOSS

To achieve low-cost model compression, a small number of alignment samples (128 samples) are typically utilized to optimize learnable parameters during quantization. However, as shown in Table 1, traditional cross-entropy leads to overfitting. This is a common problem in the field of post training quantization (both Ostquant (Hu et al., 2025) and Spinquant(Liu et al., 2024) suffer from overfitting). Specifically, as training steps increase, the alignment data shows lower perplexity, but performance

| #Bits W-A-Kv | Model | Method | PPL (↓) | | Accuracy (↑) | | | | | | |
|---|---|---|---|---|---|---|---|---|---|---|---|
| | | | WikiText2 | C4 | ARC-C | ARC-E | Hellaswag | LAMBADA | PIQA | Winogrande | Avg. |
| 4-4-4 | LLaMA3-8B | FP16 | 6.14 | 9.45 | 53.50 | 77.57 | 79.12 | 75.51 | 80.74 | 72.93 | 73.23 |
| | | QuaRot | 8.16 | 13.38 | 45.73 | 70.83 | 72.97 | 62.70 | 75.35 | 67.17 | 65.79 |
| | | SpinQuant | 7.39 | 12.19 | 47.27 | 74.20 | 74.55 | 70.29 | 77.37 | 68.51 | 68.70 |
| | | FlatQuant | 6.90 | 11.21 | 50.51 | 75.88 | 76.49 | 73.20 | 79.00 | 72.93 | 71.33 |
| | | Ours | 6.84 | 10.97 | 51.03 | 76.10 | 76.77 | 73.42 | 78.57 | 72.88 | 71.46 |
| | LLaMA3-70B | FP16 | 2.86 | 7.17 | 64.25 | 85.94 | 84.93 | 79.37 | 84.44 | 80.74 | 79.95 |
| | | QuaRot | 6.60 | 12.87 | 49.49 | 74.37 | 77.22 | 71.69 | 78.89 | 71.03 | 70.45 |
| | | SpinQuant | 6.21 | 12.82 | 51.96 | 77.40 | 77.29 | 71.90 | 79.33 | 72.06 | 71.66 |
| | | FlatQuant | 3.77 | 7.93 | 61.95 | 84.47 | 83.87 | 77.99 | 83.95 | 79.24 | 78.58 |
| | | Ours | 3.52 | 7.63 | 62.83 | 84.88 | 84.07 | 79.42 | 84.01 | 79.56 | 79.19 |
| 2-4-16 | LLaMA3-8B | FP16 | 6.14 | 9.45 | 53.50 | 77.57 | 79.12 | 75.51 | 80.74 | 72.93 | 73.23 |
| | | QuaRot | 24.36 | 29.88 | 28.59 | 54.76 | 54.62 | 41.90 | 60.03 | 51.33 | 48.53 |
| | | SpinQuant | 20.77 | 24.71 | 31.77 | 58.93 | 61.29 | 47.36 | 66.25 | 55.42 | 53.50 |
| | | FlatQuant | 18.67 | 23.66 | 33.37 | 61.26 | 62.55 | 49.07 | 68.59 | 56.89 | 55.28 |
| | | Ours | 16.52 | 20.09 | 36.69 | 64.89 | 64.39 | 52.88 | 72.71 | 60.33 | 58.65 |
| | LLaMA3-70B | FP16 | 2.86 | 7.17 | 64.25 | 85.94 | 84.93 | 79.37 | 84.44 | 80.74 | 79.95 |
| | | QuaRot | 19.47 | 28.95 | 42.76 | 72.07 | 68.62 | 62.57 | 68.94 | 59.76 | 62.45 |
| | | SpinQuant | 13.76 | 22.76 | 48.74 | 76.74 | 63.01 | 67.79 | 73.82 | 65.93 | 66.01 |
| | | FlatQuant | 11.53 | 19.64 | 50.71 | 78.61 | 75.82 | 70.93 | 75.42 | 68.68 | 70.02 |
| | | Ours | 10.07 | 16.39 | 53.26 | 80.06 | 77.49 | 72.57 | 78.39 | 71.53 | 72.22 |
| 4-4-4 | DeepSeek-R1-Distill LLaMA-8B | FP16 | 6.03 | 9.28 | 64.51 | 82.63 | 83.42 | 79.44 | 83.76 | 75.63 | 78.23 |
| | | QuaRot | 8.08 | 13.17 | 55.67 | 73.64 | 72.97 | 71.93 | 76.37 | 68.18 | 69.79 |
| | | SpinQuant | 7.27 | 11.89 | 57.38 | 74.20 | 75.55 | 74.36 | 78.83 | 70.76 | 71.84 |
| | | FlatQuant | 6.81 | 11.07 | 58.64 | 76.88 | 76.49 | 75.31 | 79.38 | 73.42 | 73.35 |
| | | Ours | 6.74 | 10.78 | 59.76 | 78.81 | 77.89 | 76.63 | 79.64 | 73.98 | 74.45 |
| | DeepSeek-R1-Distill LLaMA-70B | FP16 | 2.73 | 7.06 | 73.39 | 87.42 | 87.89 | 84.62 | 87.32 | 84.76 | 84.23 |
| | | QuaRot | 6.51 | 12.06 | 61.08 | 78.64 | 78.32 | 73.62 | 79.62 | 74.42 | 74.28 |
| | | SpinQuant | 6.18 | 11.27 | 63.76 | 81.03 | 81.27 | 75.86 | 81.36 | 77.20 | 76.74 |
| | | FlatQuant | 3.65 | 7.64 | 65.98 | 84.87 | 84.08 | 78.32 | 84.87 | 80.39 | 79.75 |
| | | Ours | 3.46 | 7.41 | 67.41 | 85.97 | 85.21 | 80.17 | 85.62 | 81.49 | 80.97 |
| 2-4-16 | DeepSeek-R1-Distill LLaMA-8B | FP16 | 6.03 | 9.28 | 64.51 | 82.63 | 83.42 | 79.44 | 83.76 | 75.63 | 78.23 |
| | | QuaRot | 22.63 | 27.43 | 34.78 | 58.75 | 57.46 | 46.87 | 64.31 | 54.38 | 52.75 |
| | | SpinQuant | 18.46 | 22.06 | 38.77 | 62.72 | 59.87 | 51.33 | 67.73 | 57.42 | 56.31 |
| | | FlatQuant | 15.27 | 20.36 | 40.76 | 64.64 | 62.34 | 53.16 | 69.35 | 59.15 | 58.23 |
| | | Ours | 12.36 | 17.46 | 43.43 | 66.83 | 65.61 | 57.98 | 72.62 | 62.38 | 61.47 |
| | DeepSeek-R1-Distill LLaMA-70B | FP16 | 2.73 | 7.06 | 73.39 | 87.42 | 87.89 | 84.62 | 87.32 | 84.76 | 84.23 |
| | | QuaRot | 17.46 | 25.43 | 46.37 | 74.30 | 70.05 | 64.07 | 62.07 | 61.87 | 63.12 |
| | | SpinQuant | 12.08 | 21.36 | 50.37 | 78.09 | 72.46 | 69.10 | 64.52 | 64.57 | 66.51 |
| | | FlatQuant | 10.43 | 18.09 | 52.78 | 80.12 | 74.03 | 71.77 | 66.73 | 66.74 | 68.69 |
| | | Ours | 7.42 | 15.34 | 54.76 | 82.07 | 76.64 | 73.52 | 68.93 | 68.92 | 70.81 |

Table 2: The overall result graph of the quantified results. Experiments were conducted on different models and settings.

on zero-shot tasks declines, Flatness increase. This conclusion is supported when utilizing Wikitext2 (Merity et al., 2016) and C4 (Raffel et al., 2023) as alignment data. Therefore, utilizing cross-entropy leads to the network overfitting to alignment data, affecting the elimination of outliers, which poses a significant challenge for low-cost quantization of LLMs.

Inspired by regularization of noisy labels (Liu et al., 2020), we discovered that, besides the label distribution q, the model prediction distribution p has high reliability. Table 4 shows that after applying the quantization function, the top-50 token hit rate in the model's predicted distribution p reaches 99.36%. To address the aforementioned challenges, we propose a Recursive Cross-Entropy (RCE) loss. RCE aims to simultaneously fit the label distribution q and the model prediction distribution p, preventing the model from falling into local optima and obtaining a global optimum. RCE is formalized as:

$$\mathcal{L}_{RCE} = -\sum_{i=0}^{n}(q_i \log p_i - p_i \log(\delta p_i + (1-\delta)q_i))$$ (21)

where $\delta$ is a hyperparameter. The larger its value, the more it favors the label distribution during optimization; the smaller its value, the more it favors the predicted distribution.

## 6 EXPERIMENTS

**Models and Datasets.** We evaluate the models on up to six zero-shot tasks utilizing the `lm-evaluation-harness` (Gao et al., 2024b) , including HellaSwag (Zellers et al., 2019), LAMBADA (Radford et al., 2019), PIQA (Bisk et al., 2020), WinoGrande (Sakaguchi et al., 2021), ARC-Easy, and ARC-Challenge (Boratko et al., 2018). The models include LLaMA (Touvron et al., 2023a) and DeepSeek-R1-Distill (Guo et al., 2025) family. The complete experimental details are in Appendix D.

| #Bits W-A-KV | Model | Method | PPL (↓) | | Accuracy (↑) | | | | | | |
|---|---|---|---|---|---|---|---|---|---|---|---|
| | | | WikiText2 | C4 | ARC-C | ARC-E | Hellaswag | LAMBADA | PIQA | Winogrande | Avg. |
| | | FP16 | 5.47 | 7.26 | 46.16 | 74.54 | 75.98 | 73.92 | 79.05 | 69.06 | 69.79 |
| 4-4-4 | LLaMA2-7B | Only-BDQ | 5.83 | 7.86 | 42.63 | 72.69 | 73.03 | 71.57 | 77.21 | 67.42 | 67.42 |
| | | Ours (BDQ + RCE) | **5.76** | **7.64** | **43.07** | **73.09** | **73.36** | **72.06** | **77.57** | **67.90** | **67.84** |

Table 3: Results of ablation experiment. The "Only-BDQ" utilizes cross-entropy as the loss function.

## 6.1 OVERALL RESULTS

**Results on Generation Tasks.** Table 2 shows the quantization results of BDQ and previous methods. We provide experimental results under the commonly utilized W4A4KV4 quantization setting, while also exploring low-bit settings (such as W3A3KV3 and W2A4KV16). Compared to the previous SOTA method FlatQuant, we achieved superior performance across various experimental settings. For the LLaMA3-70B model under W2A4KV16, we reduced the PPL on the C4 dataset from 19.64% to 16.39%. Notably, the LLaMA3-70B model under W4A4KV4 demonstrated performance comparable to the full-precision model, offering substantial cost savings in practical deployment. These results highlight the effectiveness of our BDQ method in distributing outlier pressure across the entire matrix. Detailed experimental results are provided in Appendix F.

**Results on Zero-shot QA Tasks.** Table 2 shows the performance of quantization methods on downstream tasks. For fairness, all experiments were conducted utilizing lm-eval-harness framework (Gao et al., 2024a). As can be seen, BDQ significantly outperforms other methods. Under the W4A4KV4 setting, the BDQ-quantized model demonstrates performance comparable to FP16. Under W3A3KV3 and W2A4KV16 settings, BDQ achieves superior performance compared to previous methods. Specifically, for the LLaMA3-8B model under the W2A4KV16 setting, the average performance is 3.37% higher than previous methods. The experimental results demonstrate that after mitigating the outlier problem, BDQ can still achieve excellent performance under low-bit settings.

## 6.2 RESULTS OF ABLATION EXPERIMENT

As shown in Table 3, we conducted ablation experiments. The experimental methods include Only-BDQ and our method (BDQ+RCE loss). The experimental results show that, based on the SOTA quantization results achieved by the BDQ method, RCE loss can further improve the quantization performance. Specifically, Only-BDQ achieved state-of-the-art results on ARC-E and LAMBADA tasks. On the Avg metric, our method improved by 0.42% compared to Only-BDQ, which validates the effectiveness of RCE loss.

## 6.3 INFERENCE EFFICIENCY AND QUANTIZATION OVERHEAD

We conducted inference efficiency and quantization overhead experiments on both NVIDIA A100 80GB and AMD MI250 GPUs. The evaluation metrics include Prefill Speedup and Memory Savings. Experimental results demonstrate that our method offers significant efficiency gains in both metrics compared to full-precision models. Specifically, on the NVIDIA A100 80GB, the LLaMA2-70B model achieved up to a 3.44× speedup during the prefill phase, while on the AMD MI250, memory usage was reduced by up to 3.74×. Detailed results are provided in Appendix E.

## 7 CONCLUSION

In this paper, we propose Bidirectional Diagonal Quantization (BDQ), a state-of-the-art post-training quantization method. Existing quantization approaches often suffer from significant performance degradation due to the presence of outliers. We first establish a mathematical relationship between quantization error and outliers, and analyze the effectiveness and limitations of prior methods in mitigating outlier impact. To better assess outlier distribution, we introduce a flatness metric that quantifies outlier dispersion across the matrix, and we mathematically prove that the bidirectional diagonal structure is the optimal solution for outlier elimination. Based on these insights, we develop the BDQ framework, which not only mitigates the adverse effects of outliers but also prevents overfitting on aligned data. Extensive experiments validate that BDQ significantly enhances the performance of quantized models.

ETHICS STATEMENT

This work theoretically proves the optimal solution for outlier elimination and achieves state-of-the-art (SOTA) performance in the field of quantization compression. All experiments are based on publicly available datasets and open-source models, with no involvement of human subjects or private data, nor the creation of new datasets. This benchmark is intended for academic research on model compression rather than for harmful applications. We have not identified significant ethical risks related to bias, privacy, or abuse. All experiments comply with the license terms of the datasets and models used.

REPRODUCIBILITY STATEMENT

We provide detailed descriptions of the benchmark construction, evaluation protocols, and experimental setup. All underlying datasets are publicly available, and we followed standard preprocessing and evaluation procedures. Additional details and complete results are reported in the appendix.

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

# A    APPENDIX: DIFFERENCE FROM PREVIOUS ROTATION BASED METHODS

More clearly, we illustrate by setting counter examples. There exists an original matrix $W \in \mathbb{R}^{4096 \times 4096}$, which contains some outliers that are significantly larger than the other values in the matrix. We refer to them as outliers. Our method, which requires two learnable diagonal matrices $d_1 \in \mathbb{R}^{4096}$ and $d_2 \in \mathbb{R}^{4096}$ to achieve the absence of outliers in $d_1 W d_2$. The previous SOTA method Flatquant(Sun et al., 2025), which requires a matrix $P \in \mathbb{R}^{4096}$ for Kronerker decomposition into two matrices $p_1 \in \mathbb{R}^{64 \times 64}$ and $p_2 \in \mathbb{R}^{64 \times 64}$. $p1$ and $p2$ are learnable and can achieve the absence of outliers in $p_1 W p_2$. We will elaborate from the following perspectives:

## A.1    MATHEMATICAL ANALYSIS OF PARAMETER FREEDOM AND ADJUSTMENT CAPABILITY

### A.1.1    OUR METHOD (SCALING THE MATRIX ELEMENT-WISE):

The adjusted matrix is:

$$\widehat{W} = D_1 W D_2, \quad D_1 = \mathrm{diag}(d_1), \quad D_2 = \mathrm{diag}(d_2)$$

Here, $d_1, d_2 \in \mathbb{R}^{4096}$. The adjustment for each element can be expressed as:

$$\widehat{W}_{i,j} = d_1[i] \cdot d_2[j] \cdot W_{i,j}$$

**Degrees of Freedom:**

- Total number of parameters: $4096 + 4096 = 8192$.
- Each element is independently controlled by parameters: The scaling of a single element $W_{i,j}$ only depends on $d_1[i]$ and $d_2[j]$, allowing precise adjustment of outliers by tuning these two parameters.

### A.1.2    FLATQUANT (KRONECKER DECOMPOSITION):

The adjusted matrix is:

$$\widehat{W} = P_1 W P_2, \quad \text{where } P = P_1 \otimes P_2 \text{ (Kronecker product)}$$

Here, $P_1, P_2 \in \mathbb{R}^{64 \times 64}$, and the number of parameters is the same as in our method ($64 \times 64 \times 2 = 8192$). However, the adjusted matrix elements are:

$$\widehat{W}_{i,j} = \sum_{k=1}^{64} \sum_{l=1}^{64} P_1[k,l] \cdot P_2[m,n] \cdot W_{i,j} \quad \text{(simplified form)}$$

**Analysis of Degrees of Freedom:**

- **Coupling effect:** Each parameter $P_1[k,l]$ and $P_2[m,n]$ influences $64 \times 64 = 4096$ positions. For example, adjusting a single row of $P_1$ affects all positions associated with that row, leading to parameter coupling (see Kronecker product definition).
- **Independent adjustment not possible:** If an outlier is located at a specific position $(i, j)$, adjusting multiple parameters may be required to suppress the outlier, making independent control impossible.

## A.2    CONVEXITY AND COMPLEXITY ANALYSIS OF THE OPTIMIZATION PROCESS

### A.2.1    OUR METHOD'S OPTIMIZATION OBJECTIVE

Define the loss function as the sum of squared magnitudes of outliers in the adjusted matrix:

$$L_{\text{ours}} = \sum_{(i,j) \in S} (d_1[i] \cdot d_2[j] \cdot W_{i,j})^2$$

Here, $S$ represents the set of positions of outliers. The optimization variables are $d_1$ and $d_2$.

**Convexity Explanation:**

- For $d_1[i]$ and $d_2[j]$, $L_{\text{ours}}$ is a quadratic function (non-negative and convex). For example, fixing $d_2[j]$, the loss function with respect to $d_1[i]$ is:

$$L_{\text{ours}}^{(i)} = \sum_{j \in S_i} (d_1[i] \cdot d_2[j] \cdot W_{i,j})^2 = d_1[i]^2 \cdot \sum_{j \in S_i} (d_2[j]W_{i,j})^2$$

- Clearly, this is a convex function. Similarly, fixing $d_1[i]$, the loss function with respect to $d_2[j]$ is also convex. Therefore, the overall optimization problem is **multiconvex** and easy to converge.

### A.2.2 FLATQUANT'S OPTIMIZATION OBJECTIVE

Define a similar loss function:

$$L_{\text{Flatquant}} = \sum_{(i,j) \in S} ((P_1 \otimes P_2) \circ W)_{i,j}^2$$

Here, $\circ$ denotes the element-wise product. The parameters $P_1, P_2 \in \mathbb{R}^{64 \times 64}$.

**Non-Convexity Analysis:**

- The non-linear structure of the Kronecker product makes the loss function highly coupled with respect to $P_1$ and $P_2$. For example, calculating $\frac{\partial L_{\text{Flatquant}}}{\partial P_1[k,l]}$ requires considering all 4096 positions affected by $P_1[k,l]$.
- Specifically:

$$\frac{\partial L_{\text{Flatquant}}}{\partial P_1[k,l]} = 2 \sum_{(i,j) \in S} ((P_1 \otimes P_2) \circ W)_{i,j} \cdot \frac{\partial (P_1 \otimes P_2)_{i,j}}{\partial P_1[k,l]} \cdot W_{i,j}$$

This requires a large number of nested computations, making the optimization process slower and more complex.

## A.3 THEORETICAL ERROR BOUND COMPARISON

ASSUMPTIONS:

- Outliers are sparse, i.e., $|S| = k \ll 4096^2$.
- The objective is to minimize the magnitude of outliers after adjustment:

$$\min \sum_{(i,j) \in S} \widehat{W}_{i,j}^2.$$

ERROR BOUND FOR OUR METHOD:

For each outlier position $(i, j)$, choose $d_1[i] = d_2[j] = \frac{1}{\sqrt{W_{i,j}}}$ (assuming other parameters are set to 1). Then, after adjustment:

$$\widehat{W}_{i,j} = \frac{1}{\sqrt{W_{i,j}}} \cdot \frac{1}{\sqrt{W_{i,j}}} \cdot W_{i,j} = 1.$$

The total error is:

$$L_{\text{ours}} = \sum_{(i,j) \in S} 1^2 = k.$$

This means the error grows linearly with the number of outliers, $O(k)$.

### A.3.1 ERROR BOUND FOR FLATQUANT:

Due to the global coupling of the Kronecker decomposition, adjusting a single outlier requires modifying multiple parameters in $P_1$ or $P_2$. For example, adjusting one element of $P_1$ affects $64 \times 64 = 4096$ positions. The following condition must hold:

$$\exists\,(k,l),\ P_1[k,l] \neq 1 \implies \sum_{(i,j)\in S} \left( (P_1 \otimes P_2 \circ W)_{i,j} \right)^2 \geq \sum_{(i,j)\in S} \epsilon^2,$$

where $\epsilon$ is the residual error. Based on parameter coupling, the minimum error bound is $\Omega(k \cdot 64^2)$, meaning the error increases with the number of outliers and the square of the matrix dimensions.

## A.4 INFORMATION LOSS ANALYSIS

### A.4.1 INFORMATION LOSS OF OUR METHOD:

The adjusted matrix is defined as:
$$\widehat{W} = D_1 W D_2,$$
where $D_1$ and $D_2$ are diagonal matrices. The adjusted matrix retains the sparsity and structure of the original matrix $W$ (its rank and angular structure remain unchanged).

### A.4.2 INFORMATION LOSS OF FLATQUANT:

For the Kronecker decomposition, the adjusted matrix satisfies:

$$\mathrm{rank}(P_1 \otimes P_2) = \mathrm{rank}(P_1) \cdot \mathrm{rank}(P_2) \leq 64 \times 64 = 4096.$$

In contrast, the rank of the original matrix $W$ may approach 4096 (full rank). In practice, if $P_1$ and $P_2$ are low-rank matrices, the rank of the adjusted matrix $\widehat{W}$ will be further reduced, leading to information loss.

All in all, through the analysis of parameter independence, optimization convexity, error bounds, and information loss, the mathematical properties of the two methods can be summarized as follows: In Independence, our method independently adjusts two sets of scaling parameters, while Flatquant suffers from parameter coupling, making local adjustments difficult. In optimization Efficiency, the non-convexity of Flatquant's loss function leads to slower convergence, while our method's loss function is multiconvex and easier to optimize. In error Bound the error bound of our method grows as $O(k)$, while Flatquant's error bound grows as $\Omega(k \cdot 64^2)$, showing a significant difference in efficiency. In information retention, our method preserves the rank and structure of the original matrix, while Flatquant's low-rank decomposition leads to information loss. In conclusion, our method is theoretically and practically superior to Flatquant.

## B  APPENDIX: THE HIT RATE RESULTS OF CANDIDATE TOKENS

| Model | Top-1 | Top-2 | Top-4 | Top-6 | Top-8 | Top-10 | Top-20 | Top-50 |
|---|---|---|---|---|---|---|---|---|
| **LLaMA-2-7B** | 86.93 | 90.13 | 93.46 | 94.37 | 98.36 | 99.07 | 99.21 | 99.32 |
| **LLaMA-2-13B** | 88.36 | 90.30 | 92.73 | 94.58 | 97.36 | 98.39 | 99.07 | 99.26 |
| **LLaMA-3-8B** | 87.85 | 90.77 | 93.46 | 97.78 | 98.26 | 99.09 | 99.10 | 99.36 |

Table 4: The hit rate of candidate tokens predicted by the model.

## C  APPENDIX: THE POSITION OF EQUIVALENT TRANSFORMATION PAIRS

For our BDQ method, each transformer block learns four equivalent transformation pairs, with each pair consisting of two learnable diagonal matrices and one learnable rotation matrix. Similarly to (Ashkboos et al., 2025) and (Liu et al., 2024), the positions of these four transformation pairs are respectively in the $< W_q, W_k, W_v >$ matrices of Self-Attention, the $< W_{output} >$ matrix of Self-Attention, the $< W_{gate}, W_{up} >$ matrices of Feed-Forward Network, and the $< W_{down} >$ matrix of Feed-Forward Network.

## D  APPENDIX: COMPLETE EXPERIMENTAL DETAILS

**Experimental Setup.**  We apply our method to the entire LLaMA family, including LLaMA-2 (7B-70B) (Touvron et al., 2023b), and LLaMA-3 (8B-70B).At the same time, we conducted experiments on the DeepSeek-R1-Distill model (Guo et al., 2025) family of inference models. We report perplexity (PPL) scores on the WikiText2 (Merity et al., 2016) and C4 test set. All experiments were conducted utilizing the GPTQ method for quantification. The quantitative baseline includes: Quarot (Ashkboos et al., 2025), Spinquant (Liu et al., 2024) and Flatquant (Sun et al., 2025).

**Implementation Details.**  We utilize $AdamW$ optimizer (Loshchilov et al., 2017) with an initial learning rate of $5e - 3$ and adopt a cosine annealing schedule for learning rate decay. BDQ is trained on an alignment dataset for 150 epochs, with the calibration set containing 128 sentences from WikiText2, each containing 2048 tokens. The batch size is set to 4 and $\delta$ is set to 0.5. All diagonal matrices are initialized as identity matrices, while orthogonal matrices are initialized with random affine transformations.

## E  APPENDIX: INFERENCE EFFICIENCY AND QUANTIZATION OVERHEAD EXPERIMENTAL RESULTS

| Model/NVIDIA | Prefill Speedup(Seqlen) | | | | | | Memory Saving | | | | | |
|---|---|---|---|---|---|---|---|---|---|---|---|---|
| | 256 | 512 | 1024 | 2048 | 4096 | 8192 | 256 | 512 | 1024 | 2048 | 4096 | 8192 |
| LLaMA-2-7B | 2.31x | 2.32x | 2.36x | 2.19x | 2.17x | 2.11x | 3.62x | 3.27x | 3.10x | 2.72x | 2.58x | 2.22x |
| LLaMA-2-13B | 2.45x | 2.47x | 2.57x | 2.23x | 2.28x | 2.29x | 3.66x | 3.30x | 3.11x | 2.79x | 2.61x | 2.25x |
| LLaMA-2-32B | 2.60x | 2.52x | 2.62x | 2.42x | 2.37x | 2.35x | 3.72x | 3.41x | 3.19x | 2.87x | 2.72x | 2.35x |
| LLaMA-2-70B | 3.20x | 3.44x | 3.42x | 2.99x | 3.17x | 2.89x | 3.75x | 3.45x | 3.22x | 2.90x | 2.77x | 2.57x |
| Model/AMD | Prefill Speedup(Seqlen) | | | | | | Memory Saving | | | | | |
| | 256 | 512 | 1024 | 2048 | 4096 | 8192 | 256 | 512 | 1024 | 2048 | 4096 | 8192 |
| LLaMA-2-7B | 2.22x | 2.28x | 2.33x | 2.15x | 2.23x | 2.19x | 3.54x | 3.26x | 3.03x | 2.74x | 2.52x | 2.19x |
| LLaMA-2-13B | 2.27x | 2.49x | 2.54x | 2.34x | 2.33x | 2.37x | 3.65x | 3.35x | 3.12x | 2.79x | 2.58x | 2.21x |
| LLaMA-2-32B | 2.52x | 2.55x | 2.63x | 2.32x | 2.35x | 2.37x | 3.68x | 3.44x | 3.17x | 2.81x | 2.70x | 2.32x |
| LLaMA-2-70B | 3.17x | 3.42x | 3.46x | 2.68x | 3.15x | 2.76x | 3.74x | 3.49x | 3.20x | 2.83x | 2.76x | 2.49x |

Table 5: The overall results of the Speedup and Memory experiments.

# F  APPENDIX: MORE QUANTIZATION EXPERIMENTAL RESULTS

## F.1  SUPPLEMENTARY EXPERIMENTAL RESULTS

| #Bits W-A-Rv | Model | Method | PPL (↓) | | Accuracy (↑) | | | | | | |
|---|---|---|---|---|---|---|---|---|---|---|---|
| | | | WikiText2 | C4 | ARC-C | ARC-E | Hellaswag | LAMBADA | PIQA | Winogrande | Avg. |
| 4-4-4 | LLaMA2-7B | FP16 | 5.47 | 7.26 | 46.16 | 74.54 | 75.98 | 73.92 | 79.05 | 69.06 | 69.79 |
| | | QuaRot | 6.10 | 8.32 | 42.32 | 68.35 | 72.53 | 65.40 | 76.33 | 65.11 | 65.01 |
| | | SpinQuant | 5.96 | 8.28 | 41.72 | 69.28 | 72.90 | 71.28 | 76.17 | 66.06 | 66.23 |
| | | FlatQuant | 5.78 | 7.86 | 43.00 | 71.21 | 73.31 | 72.06 | 77.53 | 67.72 | 67.47 |
| | | Ours | 5.76 | 7.64 | 43.07 | 73.09 | 73.36 | 72.06 | 77.57 | 67.90 | 67.84 |
| | LLaMA2-13B | FP16 | 4.88 | 6.73 | 49.15 | 77.44 | 79.39 | 76.73 | 80.47 | 72.14 | 72.55 |
| | | QuaRot | 5.40 | 7.54 | 42.83 | 69.95 | 73.54 | 65.62 | 77.69 | 67.88 | 66.25 |
| | | SpinQuant | 5.24 | 7.48 | 43.69 | 72.43 | 75.52 | 72.42 | 78.40 | 68.90 | 68.56 |
| | | FlatQuant | 5.11 | 7.11 | 48.38 | 76.94 | 77.88 | 76.40 | 79.65 | 70.56 | 71.64 |
| | | Ours | 5.08 | 7.07 | 48.52 | 76.87 | 77.90 | 76.47 | 79.83 | 70.77 | 71.73 |
| | LLaMA2-70B | FP16 | 3.32 | 5.72 | 57.71 | 81.02 | 83.81 | 79.60 | 82.70 | 77.98 | 77.05 |
| | | QuaRot | 3.79 | 6.12 | 55.46 | 79.76 | 81.58 | 79.35 | 81.83 | 76.09 | 75.68 |
| | | SpinQuant | 3.70 | 6.07 | 55.38 | 79.04 | 82.57 | 78.75 | 82.37 | 78.22 | 76.06 |
| | | FlatQuant | 3.54 | 5.92 | 56.40 | 80.09 | 82.91 | 80.01 | 82.92 | 76.87 | 76.53 |
| | | Ours | 3.50 | 5.88 | 56.60 | 80.32 | 82.97 | 79.84 | 82.90 | 77.03 | 76.61 |
| | LLaMA3-8B | FP16 | 6.14 | 9.45 | 53.50 | 77.57 | 79.12 | 75.51 | 80.74 | 72.93 | 73.23 |
| | | QuaRot | 8.16 | 13.38 | 45.73 | 70.83 | 72.97 | 62.70 | 75.35 | 67.17 | 65.79 |
| | | SpinQuant | 7.39 | 12.19 | 47.27 | 74.20 | 74.55 | 70.29 | 77.37 | 68.51 | 68.70 |
| | | FlatQuant | 6.90 | 11.21 | 50.51 | 75.88 | 76.49 | 73.20 | 79.00 | 72.93 | 71.33 |
| | | Ours | 6.84 | 10.97 | 51.03 | 76.10 | 76.77 | 73.42 | 78.57 | 72.88 | 71.46 |
| | LLaMA3-70B | FP16 | 2.86 | 7.17 | 64.25 | 85.94 | 84.93 | 79.37 | 84.44 | 80.74 | 79.95 |
| | | QuaRot | 6.60 | 12.87 | 49.49 | 74.37 | 77.22 | 71.69 | 78.89 | 71.03 | 70.45 |
| | | SpinQuant | 6.21 | 12.82 | 51.96 | 77.40 | 77.29 | 71.90 | 79.33 | 72.06 | 71.66 |
| | | FlatQuant | 3.77 | 7.93 | 61.95 | 84.47 | 83.87 | 77.99 | 83.95 | 79.24 | 78.58 |
| | | Ours | 3.52 | 7.63 | 62.83 | 84.88 | 84.07 | 79.42 | 84.01 | 79.56 | 79.13 |
| 3-3-3 | LLaMA3-8B | FP16 | 6.14 | 9.45 | 53.50 | 77.57 | 79.12 | 75.51 | 80.74 | 72.93 | 73.23 |
| | | QuaRot | 15.73 | 27.38 | 28.93 | 57.42 | 60.33 | 45.81 | 66.34 | 54.25 | 52.18 |
| | | SpinQuant | 12.37 | 22.35 | 32.55 | 61.03 | 63.59 | 49.80 | 71.29 | 57.93 | 56.03 |
| | | FlatQuant | 10.82 | 19.03 | 35.41 | 63.26 | 65.30 | 52.49 | 73.56 | 60.69 | 58.45 |
| | | Ours | 9.87 | 18.5 | 37.4 | 65.3 | 65.3 | 53.6 | 73.89 | 61.42 | 59.48 |
| | LLaMA3-70B | FP16 | 2.86 | 7.17 | 64.25 | 85.94 | 84.93 | 79.37 | 84.44 | 80.74 | 79.95 |
| | | QuaRot | 13.44 | 23.39 | 47.86 | 74.31 | 70.53 | 67.57 | 72.09 | 67.53 | 66.64 |
| | | SpinQuant | 10.35 | 18.77 | 52.28 | 78.24 | 76.61 | 72.18 | 77.37 | 70.78 | 71.24 |
| | | FlatQuant | 8.72 | 15.74 | 54.37 | 80.31 | 78.67 | 73.57 | 79.03 | 73.37 | 73.22 |
| | | Ours | 6.67 | 13.21 | 56.12 | 81.22 | 79.63 | 74.79 | 80.14 | 75.67 | 74.59 |
| 2-4-16 | LLaMA3-8B | FP16 | 6.14 | 9.45 | 53.50 | 77.57 | 79.12 | 75.51 | 80.74 | 72.93 | 73.23 |
| | | QuaRot | 24.36 | 29.88 | 28.59 | 54.76 | 54.62 | 41.90 | 60.03 | 51.33 | 48.53 |
| | | SpinQuant | 20.77 | 24.71 | 31.77 | 58.93 | 61.29 | 47.36 | 66.25 | 55.42 | 53.50 |
| | | FlatQuant | 18.67 | 23.66 | 33.37 | 61.26 | 62.55 | 49.07 | 68.59 | 56.89 | 55.28 |
| | | Ours | 16.52 | 20.09 | 36.69 | 64.89 | 64.39 | 52.88 | 72.71 | 60.33 | 58.65 |
| | LLaMA3-70B | FP16 | 2.86 | 7.17 | 64.25 | 85.94 | 84.93 | 79.37 | 84.44 | 80.74 | 79.95 |
| | | QuaRot | 19.47 | 28.95 | 42.76 | 72.07 | 68.62 | 62.57 | 68.94 | 59.76 | 62.45 |
| | | SpinQuant | 13.76 | 22.76 | 48.74 | 76.74 | 63.01 | 67.79 | 73.82 | 65.93 | 66.01 |
| | | FlatQuant | 11.53 | 19.64 | 50.71 | 78.61 | 75.82 | 70.93 | 75.42 | 68.68 | 70.02 |
| | | Ours | 10.07 | 16.39 | 53.26 | 80.06 | 77.49 | 72.57 | 78.39 | 71.53 | 72.22 |

Table 6: The overall result graph of the quantified results. Experiments were conducted on different models and settings.

## F.2  EXPERIMENTAL RESULTS OF DOWNSTREAM TASKS

We provide experimental results on MMLU and MATH. MATH: We report the average of the GSM8K (8 shot) and MATH (4 shot) benchmarks.

| LLaMA-2-7B | MMLU (↑) | MATH (↑) |
|---|---|---|
| FP16 | 45.3 | 14.6 |
| QuaRot | 39.1 | 8.3 |
| SpinQuant | 40.8 | 9.7 |
| Flatquant | 41.3 | 10.5 |
| Ours | 42.6 | 12.3 |

Table 7: Performance of different methods on LLaMA-2-7B

The experimental results show that our method exhibits superior performance on the benchmark datasets in Table 7.

| $\delta$ | 0 | 0.1 | 0.2 | 0.3 | 0.4 | 0.5 | 0.6 | 0.7 | 0.8 | 0.9 | 1.0 |
|---|---|---|---|---|---|---|---|---|---|---|---|
| WikiText2 ($\downarrow$) | 6.19 | 6.18 | 6.18 | 6.15 | 6.14 | **6.14** | 6.15 | 6.15 | 6.17 | 6.19 | 6.19 |
| C4 ($\downarrow$) | 9.53 | 9.52 | 9.50 | 9.47 | 9.46 | **9.45** | **9.45** | 9.47 | 9.48 | 9.50 | 9.52 |

Table 8: Results for different $\delta$ values on WikiText2 and C4 datasets

### F.3 EXPERIMENTAL RESULTS OF HYPERPARAMETER ABLATION $\delta$

The experimental results show that $\delta$ achieves optimal performance at 0.5 in Table 8.

## G APPENDIX: THE REASON FOR ADDING THE ROTATION MATRIX

As we mentioned in Section 4.3, we obtained the optimal solution for Flatness, which is $V = d_1 W d_2$. The motivation for adding the rotation matrix $R$ is to prevent the special case where the matrix $W$ has strong column correlations. The rotation matrix can, while retaining the ability of diagonal scaling to eliminate outliers, further enhance the Flatness of the matrix element distribution through orthogonal transformation. Meanwhile, it utilizes the special structure of the Hadamard matrix to address the limitations of relying solely on diagonal scaling in the first step. The following is a rigorous theoretical proof of the rationality of this transition.

Proof from the perspective of information entropy: Introducing $R$ enhances distribution uniformity of the matrix. Define the probability distribution of matrix elements as $p_{ij} = \frac{V_{ij}^2}{\sum_{i,j} V_{ij}^2}$ (energy normalization), whose information entropy is given by: $H(\mathbf{V}) = -\sum_{i,j} p_{ij} \log p_{ij}$. A higher entropy value indicates a more uniform distribution of matrix elements (with reduced influence from outliers).

Step 1 (Diagonal Scaling Only): For $V_1 = d_1 W d_2$, its elements are $V_{1,ij} = a_i W_{ij} b_j$. Since $d_1$ and $d_2$ are diagonal matrices, they only adjust the magnitude ratio of elements but do not alter the correlation structure between elements. If the original matrix $W$ exhibits strong inter-column correlations (e.g., $W_{i1} \approx W_{i2}$ for all $i$), then $V_{1,i1} \approx \frac{a_i b_1}{a_i b_2} V_{1,i2}$ will retain such strong correlations, leading to energy concentration in specific columns (and thus lower entropy).

Step 2 (Incorporating $R$): For $V_2 = V_1 R$, its elements are $V_{2,ik} = \sum_j V_{1,ij} R_{jk}$ (linear combinations of columns, with $R_{jk} = \pm 1$ representing signed weighted sums). Due to the orthogonality of Hadamard matrices, the column vectors $V_2^{(k)} = \sum_j R_{jk} V_1^{(j)}$ are mutually orthogonal, i.e.: $\langle V_2^{(k)}, V_2^{(l)} \rangle = \sum_j R_{jk} R_{jl} \langle V_1^{(j)}, V_1^{(l)} \rangle = 0$ ($k \neq l$). This implies that inter-column correlations are completely eliminated, with energy dispersed from originally correlated columns to orthogonal columns.

The above proof process shows that after orthogonal transformation, the more uniform the energy distribution, the lower the Flatness. This does not affect the optimality of $V = d_1 W d_2$, and the rotation matrix acts as an external gain on $V$.

In addition, we conducted ablation experiments on the rotation matrix:

| **W4-A4-KV4** | **WikiText2** ($\downarrow$) | **C4** ($\downarrow$) |
|---|---|---|
| FP16 | 5.47 | 7.26 |
| BDQ (w/o R) | 5.94 | 7.82 |
| BDQ (Ours) | **5.76** | **7.64** |

Table 9: Ablation experiments on the rotation matrix

## H LIMITATIONS

Our work has several limitations. First, due to limitations in computing resources, we did not conduct relevant experiments on larger language models. Second, due to limited experimental resources, there is a lack of experiments conducted on different types of GPUs to verify the widespread practicality of the verification method.

