# OpenReview forum: "BDQ: Bidirectional Diagonal Quantization for LLMs"
_ICLR.cc/2026/Conference — ICLR 2026 Conference Withdrawn Submission_

### Official Review · Reviewer_KjKD · 2025-10-24

**Soundness:** 2
**Presentation:** 2
**Contribution:** 2
**Rating:** 2
**Confidence:** 5

**Summary:**

This work proposed BDQ, called bi-directional diagonal quantization for LLMs. It aims to optimize two diagonal matrices, which are scaling factors on both input channels and output channels. The authors first give a motivation section on outliers, and then demonstrate the necessity of separating column-wise and row-wise outliers.

**Strengths:**

+ The definition of Flatness is interesting, but I did not get why $\alpha_i$ and $\beta_i$ has to be defined like this. The current formula looks like backtracking from the methodology rather than straightforwardly leading to the optimization method.

+ There is a reasoning model evaluation, but I think a more concrete evaluation is the actual Reasoning tasks as listed in DeepSeek-Distill model card page.

**Weaknesses:**

I have several concerns about this work; therefore, I have given the rejection right now. If the authors can clarify and convince me, I will reconsider my scores.

## Major Weakness

+ Section 5.1 and Appendix C does not provide enough detail (and there is no code implementation). I am curious about how the scaling matrix Eq. 19 can be eliminated. Based on the current form, I cannot imagine how $\Lambda_1$ is applied to $x$. It cannot a scaling on the batch dimension, cause that would be dynamic. Also, it cannot be a scaling on the channel dimension because $\Lambda_2$ should be on the channel dimension and will be canceled off with the $\Lambda_2^{-1}$ on the weights. The authors should clearly explain how the dimensions of this matrix are defined. In the meantime, the diagonal matrix seems to be computed on the fly rather than fused into other modules, which adds quantization overhead.

+ The authors spent so much effort to illustrate the importance of diagonal matrices. First of all, I don't think this is an elegant design, and it still relies on the Hadamard transform matrix, which makes me curious about its performance without the Hadamard transform.

+ The bi-directional diagonal or two-dimensional scaling methods are not motivated properly. For the whole sections 3 and 4, this work discusses the weight outlier and leaves activation aside. While in the actual methodology part, it seems like the activation outliers should be considered as well, and the authors choose a global optimization method, which is completely irrelevant to Sections 3 and 4.

+ Section 3 analysis is not rigorous; there are no activation outliers; it just considers the average effect over a whole matrix; the final conclusion in Eq. 10 is so primitive that it can only be used as an upper bound rather than the actual outlier effect. Meanwhile, it also has no connection with the global loss objective.

## Minor Weakness

+ For DeepSeek-Distilled models, I would prefer to look at the reasoning tasks' accuracy rather than these tasks. I assume they are evaluated only with log-likelihood rather than performing the decoding of a reasoning question.

+ No hardware overhead is discussed for applying BDQ in reality.

**Questions:**

How is Eq. 21 RECURSIVE? It looks like an interpolation rather than a recursion.

---

### Official Review · Reviewer_QTwa · 2025-10-29

**Soundness:** 2
**Presentation:** 1
**Contribution:** 1
**Rating:** 4
**Confidence:** 4

**Summary:**

The paper analyzes quantization error in the presence of outliers in LLMs and proposes an entropy-inspired metric, Flatness, to quantify distribution uniformity. It derives an diagonal scaling form and introduces Bidirectional Diagonal Quantization (BDQ) with a Recursive Cross-Entropy (RCE) loss to reduce calibration overfitting. Experiments report strong results across LLaMA families.

**Strengths:**

1. The paper contains certain part of theoretical analysis.
2. The proposed method is assessed through a variety of experiments.
3. The proposed metric, flatness, makes sense and works.

**Weaknesses:**

1. The **bold notation is problematic** in Table 2: For LLaMA3-8B W4A4KV4, the proposed method doesn’t achieve the best performance on PIQA and WINOGRANDE, yet it is bolded. The same issue appears for: (1) LLaMA2-7B W4A4KV4, LAMBADA; (2) LLaMA2-13B W4A4KV4, ARC-E; (3) LLaMA2-70B W4A4KV4, LAMBADA/Winogrande/PIQA. The bolding problem also occurs in Table 8. In addition, for LLaMA3-8B W3A3KV3, the proposed method should report at least two decimal places to align with baselines.

2. There is a contradiction between Tables 2 and 6. For LLaMA3-70B, the proposed method shows different average QA scores across the two tables. It is also odd that **four identical settings** appear in both tables.

3. Some statements are sloppy. At Line 171, the authors assume the error follows a normal distribution, but Eq. (7) uses a **uniform** quantization-noise model. Also, $\Delta'$ is unclear: from Eq. (4) we get $\Delta' = |w_{\text{outlier}}|/(2^b-1)$, but later we see $\Delta' \ll |w_{\text{outlier}}|/(2^b-1)$. Overall, the theory in Sec. 3 is messy and inconsistent.

4. It remains unclear how $R$ in Eq. (19) is obtained. What exactly is meant by “the rotation matrix $R$ is composed of a Hadamard matrix and an additional orthogonal matrix”? How is the orthogonal matrix optimized? Also, the procedure for obtaining the results in Table 1 is not described.

5. There is no detailed illustration of the **hardware/inference implementation** that yields efficiency. The paper provides tables and numbers but no kernel/inference implementation details. Moreover, the title of Sec. 6.3 mentions **overhead**, but no corresponding results or analysis are given.

6. From my point of view, Sec. 3 should not be titled “Motivation.” That outliers make quantization challenging is well known, and smoothing operations already exist in prior work. Many methods use $W d$ rather than $d_1 W d_2$ because outliers are typically channel-wise/token-wise. The motivation should explain **why bi-directional smoothing is better**, not simply reiterate that outliers impair quantization.

7. Novelty is moderate. The method appears incremental relative to rotation-based approaches such as SpinQuant/SmoothQuant.

8. There is no quantization runtime (not just inference efficiency) comparison to other baselines. Will the proposed objective over the whole model be slower than layer-wise optimization?

9. Please add an empirical ablation demonstrating the effectiveness of using **bidirectional** diagonal matrices versus using only a single diagonal matrix.

**Questions:**

See weaknesses above.

---

### Official Review · Reviewer_TaMt · 2025-10-31

**Soundness:** 3
**Presentation:** 3
**Contribution:** 3
**Rating:** 4
**Confidence:** 2

**Summary:**

This paper studies PTQ for LLMs via a principled route: (1) it derives that quantization error grows quadratically with outlier magnitude; (2) it proposes an entropy-inspired Flatness to measure outlier dispersion; (3) it proves that bidirectional diagonal scaling is optimal for Flatness; and (4) it instantiates BDQ, which inserts equivalent diagonal-plus-rotation pairs to disperse outliers without changing FP outputs. A Recursive Cross-Entropy further mitigates overfitting to tiny calibration sets.

**Strengths:**

1. Clear theory-to-method pipeline; optimal-form result is elegant and actionable.

2. Extensive results on LLaMA/DeepSeek-R1 show strong gains over QuaRot/SpinQuant/FlatQuant under W4A4 and W2A4KV16.

**Weaknesses:**

1. Proofs assume specific constraints (energy normalization, diagonal family). Could the authors formalize limits vs. more general linear transforms? Any counter-examples?

2. Normal-error assumption and probability-style treatment of $𝑊^2/(𝛼𝛽)$ deserve deeper justification or sensitivity checks.

3. The paper would benefit from a side-by-side comparison with DuQuant [1] under matched settings—same checkpoints, KV precision, group sizes, calibration set size, evaluation harness/version, and decoding setup.

[1] Lin et al. DuQuant: Distributing Outliers via Dual Transformation Makes Stronger Quantized LLMs. NeurIPS 2024.

**Questions:**

Please see the Weakness.

1. Can the authors provide kernel-level/runtime analysis (overheads of diagonal/rotation, cache behavior, parallelism) beyond the appendix summaries? What is the precision used to compare with the full precision model in the article's inference efficiency test?

---

### Official Review · Reviewer_XcHX · 2025-11-08

**Soundness:** 2
**Presentation:** 2
**Contribution:** 2
**Rating:** 2
**Confidence:** 4

**Summary:**

The paper introduces Bidirectional Diagonal Quantization (BDQ), a post-training quantization method that models the mathematical link between outliers and quantization error, defines a Flatness metric to measure outlier dispersion, and proposes row/column diagonal scaling with a Hadamard rotation to optimize Flatness. A Recursive Cross-Entropy loss is further applied to mitigate overfitting when using small calibration datasets. Extensive experiments are reported across multiple LLaMA and DeepSeek models, showing improvements over prior PTQ baselines.

**Strengths:**

- Clear motivation.
- Lightweight parameterization (two diagonals + orthogonal rotation) that integrates smoothly with PTQ pipelines.
- Consistent empirical improvements on multiple models under W4A4/W2A4 settings.

**Weaknesses:**

1. **Limited novelty — overlaps with established PTQ theory.** The claimed “analytical connection between outlier magnitude and quantization error” (Sec. 3, Eq. 5–10) is not conceptually new. Prior PTQ works such as SmoothQuant, AWQ, OmniQuant, and QuaRot/SpinQuant already formalized that large activations enlarge the quantization scale and induce clipping-dominated MSE.
Even the FlatQuant (ICML2025) paper—explicitly cited here—uses a *flatness* notion to handle outliers. BDQ’s theoretical component mainly restates well-known quantization principles: it revisits the classic relationship between outlier magnitude, quantization scale expansion, and error growth, presenting it with cleaner notation but no fundamentally new insight. The claimed “optimal” diagonal transformation is conceptually similar to long-established row/column scaling strategies used in PTQ to balance activation ranges.
However, the paper treats this as a unique and theoretically derived solution, while the formulation still allows many equivalent scalings that produce identical results. As such, the theoretical contribution feels more like a refined exposition of existing ideas than a genuine advance in quantization theory.

2. **Evaluation focuses on easy, saturated benchmarks.** Most reported gains come from HellaSwag, ARC-E/C, PIQA, WinoGrande, LAMBADA—tasks where modern LLMs already perform near ceiling and quantization rarely breaks performance. These benchmarks measure shallow commonsense recall, not the reasoning or compositional precision that stresses quantized arithmetic. Crucially, the paper evaluates MMLU, GSM8K, MATH only on LLaMA-2-7B, not on the stronger LLaMA-3 or DeepSeek-R1-Distill models that underpin the main claims. Without results on such hard, discriminative benchmarks, the empirical evidence for robustness under quantization remains weak.

3. **Overstated performance claims.** The abstract’s “ <1% accuracy drop on LLaMA-3-8B W4A4 ” conflicts with Table 2 likewise, the “ 39.1% gap reduction ” lacks a consistent metric definition. These inconsistencies suggest selective interpretation of numbers rather than a holistic evaluation.

4. **Theory–practice gap.** While the Flatness metric is motivated by entropy, experiments never correlate measured Flatness reduction with accuracy retention on challenging tasks. The connection between the theoretical objective and actual quantization quality is thus anecdotal.

**Questions:**

Covered at Weaknesses section.

---

### Note · Authors · 2025-12-26

I have read and agree with the venue's withdrawal policy on behalf of myself and my co-authors.